# QUANTUM FOURIER NETWORKS FOR SOLVING PARAMETRIC PDES

## ABSTRACT

Many real-world problems like modelling environment dynamics, physical processes, time series etc., involve solving Partial Differential Equations (PDEs) parameterized by problem-specific conditions. Recently, a deep learning architecture called Fourier Neural Operator (FNO) proved to be capable of learning solutions of given PDE families, for any initial conditions as input. Given the advancements in quantum hardware and the recent results in quantum machine learning methods, we propose three quantum circuits, inspired by the FNO, to learn this functional mapping for PDEs. The proposed algorithms are distinguished based on the trade-off between depth and their similarity to the classical FNO. At their core, we make use of unary encoding paradigm and orthogonal quantum layers, and introduce a new quantum Fourier transform in the unary basis. With respect to the number of samples, our quantum algorithm is proven to be substantially faster than the classical counterpart. We benchmark our proposed algorithms on three PDE families, namely Burger's equation, Darcy's flow equation and the Navier-Stokes equation, and the results show that our quantum methods are comparable in performance to the classical FNO. We also show an analysis of the image classification tasks where our proposed algorithms are able to match the accuracy of the CNNs, thereby showing their applicability to other domains.

## 1 INTRODUCTION

Solving Partial Differential Equations (PDEs) has been a very crucial step in understanding the dynamics of nature. They have been widely used to understand natural phenomenon such as heat-transfer, modelling the flow of fluids, electromagnetism, etc. and lately have also found their applications in understanding the behavior of dynamical markets. In most of these applications, a closed form solution is difficult to find for the resulting PDEs and thus, classical solvers require a lot of evaluations to model the solution for a given PDE. To approximate such not-so-easily solvable PDE's, there has been an extensive research based on *neural networks*. A group of these methods Yu et al. (2018); Raissi et al. (2019); Bar & Sochen (2019) aimed at learning the solution function for an instance of PDE and thus requires to be re-trained every time the parameters/conditions in the PDE change. The other set of methods Zhu & Zabaras (2018); Adler & Öktem (2017); Bhatnagar et al. (2019) targeted at learning over a family of PDEs but for a specific resolution dependent, making these methods limited to the discretization or the sampling density used in the training data. A recent work Li et al. (2020) addressed both these issues and posed the problem as learning a function-to-function mapping for parametric PDEs. Experiments on widely popular PDEs showed that it was effective in learning the mapping from a parametric initial condition function to the solution operator for a family of PDEs. The method proposes a Fourier layer, which uses a learnable linear transform sandwiched between a Fourier Transform ($FT$) and an Inverse Fourier Transform ($IFT$) operation. This is similar to the convolution operation as it also translates to multiplication in the Fourier space.

The major bottleneck which might hinder the scalability of this classical Fourier Neural Operator (FNO) is its time complexity, limited by the classical Fourier Transform (FT) and Inverse Fourier Transform (IFT) operations inside the Fourier Layer. As shown previously by Musk (2020), these operations are much faster when deployed on quantum hardware. Similar advantages have led to significant developments in learning approaches based on near term quantum computing. The initial demonstrations of these algorithms involved experiments on a small-scale hardware Farhi & Neven

(2018); Coyle et al. (2020); Cappelletti et al. (2020); Grant et al. (2018) which established their effectiveness in extracting patterns. Following this, many works Abbas et al. (2021); Mari et al. (2020); Beer et al. (2020); Allcock et al. (2020) proposed small-scale implementations of fully connected quantum neural networks on near term hardware. Other proposals Kerenidis et al. (2019); Cong et al. (2019) for deploying convolution-based learning methods on quantum devices showed effective training in practice. Furthermore, Chakrabarti et al. (2019) proposed quantum-hardware implementation for generative adversarial networks. A different approach, where the inputs are encoded as unary states, using the two-qubits quantum gate RBS (*Reconfigurable Beam Splitter*) was proposed in a recent work Johri et al. (2021). This encoding gave rise to the use of orthogonal properties of pure quantum unitaries, as proposed in Kerenidis et al. (2021) for training, for instance, orthogonal feed-forward networks to damp the gradient based issues while learning. It used a pyramid circuit based on parameterized RBS gates to implement a learnable orthogonal matrix as compared to the existing classical approaches which offer approximate orthogonality at the cost of increased training time. This orthogonality in neural networks results in much smoother convergence and also lesser parameters as shown by Li et al. (2019) for feed-forward neural networks and Wang et al. (2020) for convolutional nets. The effectiveness of these orthogonal quantum networks was further shown in another work on medical image classification Mathur et al. (2021) problem. We also use a similar idea to do the multi-channel intermediate linear transform using orthogonal matrix but using parameterized butterfly circuits instead of pyramid circuits.

Exploiting these advantages offered by quantum computing, in this work, we propose a new kind of Quantum Fourier Transform (QFT), which operates on the unary states and a learnable Quantum Linear Transform. We further propose three quantum algorithms inspired by this classical Fourier operation which are faster than the classical operation and require fewer parameters for the same architecture, thereby boosting their scalability. Given the input of dimension $N_s \times N_c$, where $N_s$ corresponds to number of samples per PDE and $N_c$ correspond to feature dimension, the order of time complexity corresponding to Fourier Layer (FL) and proposed algorithms is shown in table 1.

Table 1: Comparison of order of time/depth complexities ($O$) of the proposed circuits with the existing classical Fourier Layer (FL). Here $N_s$ denote the sampling dimension, $N_c$ denote the feature dimension where $N_s \gg N_c$ and $K$ (usually in range 4-16) denotes the maximum number of modes allowed Li et al. (2020). This implies that the proposed quantum algorithms would be faster than the classical method. Each quantum circuit requires $N_c + N_s$ qubits and $K$ independent parallel circuits are required by the Parallelized QFNO.

| Method | Classical FL | Sequential Quantum FL | Parallel Quantum FL | Compound Quantum FL |
|---|---|---|---|---|
| Complexity | $N_c + N_s \log(N_s)$ | $K\log(N_c) + N_c\log(N_s)$ | $\log(N_c) + N_c\log(N_s)$ | $\log(N_c + K) + N_c\log(N_s)$ |
| # qubits | - | $N_c + N_s$ | $N_c + N_s$ | $N_c + N_s$ |
| # parallel circuits | - | 1 | $K$ | 1 |

The first algorithm replicates the classical operation on a quantum circuit. The other two algorithms are modifications of the first circuit designed for the noisy learning process offered by the near term quantum hardware. We test all the three proposed algorithms on all the three PDEs evaluated in the classical FNO paper Li et al. (2020) namely the Burgers equation, Darcy's Flow equation and Navier Stokes equation on the synthetic datasets used in that paper. We also test our algorithms against the Convolutional neural networks (CNNs) on benchmark datasets for image classification namely MNIST, Fashion-MNIST Xiao et al. (2017), Pneumonia-MNIST Yang et al. (2021). In all the experiments, three algorithms perform similarly and comparable to state-of-the-art FNO for PDEs. Also, they perform decently on the image classification tasks.

## 2 CLASSICAL FOURIER NEURAL OPERATOR

Given a training set comprising the family of a Partial Differential Equation, the classical FNO Li et al. (2020) aims to learn a functional mapping from a parameterized initial condition to the solution function for this family. This means given an initial condition function characterizing a PDE instance, sampled at different points, it should predict the solution function values at those points at the inference time. To formulate it, given two functional spaces $\mathcal{A}$ and $\mathcal{U}$ along with a set of observations $\{a_j, u_j\}$ ($a_j \sim \mu$ is an i.i.d. sequence sampled from some function $f \in A$), it learns a parametric mapping $G : \mathcal{A} \times \Theta \to \mathcal{U}$. To achieve this, it proposed a learning network based on iteratively applying a new kind of layer which it termed as the Fourier Layer. The layer consists of two parts, the top part involves firstly projecting the input to the Fourier domain and then applying

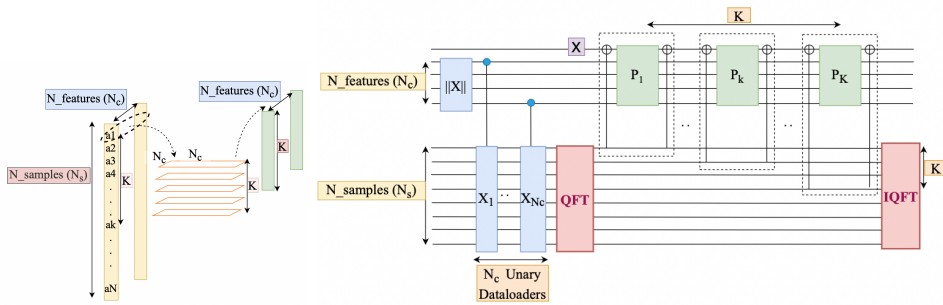

(a) **Left**: Classical Fourier Layer      (b) **Right**: Sequential Quantum Circuit for FNO.

Figure 1: **Left**: Intermediate Linear Transform operation performed in the classical FNO, after applying the Fourier Transform Li et al. (2020). As shown, it performs a linear transform on the features corresponding to the first $K$ modes in the Fourier Transformed input. **Right**: The proposed Sequential Quantum Circuit which replicates the classical FNO operation. Further details regarding it are given in Section 3.2.1. The dashed box incorporates the controlled butterfly incorporating a controlled norm-loader and the parametric butterfly ($P_k$).

a linear transform (refer fig. 1) to first $K$ modes and crop the rest and then reproject back to the original domain. This is somewhat similar to convolution but the updates would be taking place in the Fourier space. The lower part is just a simple convolution and doesn't play any significant role in improving performance. In this paper, we propose purely quantum algorithms for this top part as a *Quantum Fourier Layer* and term the network as *Quantum Fourier Neural Operator* (QFNO).

Now, we look at the mathematical details of the classical Fourier layer for a 1D PDE case (eg. Burger's equation), showing the inputs and outputs of each transformation involved. Given a classical input $x \in \mathbb{R}^{N_c \times N_s}$, where $N_c$ corresponds to the number of channels/features in the input and $N_s$ corresponds to number of samples/observations per function corresponding to a PDE, we denote the corresponding output of this classical operation by $y \in \mathbb{R}^{N_c \times N_s}$. As the quantum matrices are orthogonal and the $l_2$-norm of any quantum state vector is 1, we first normalize the input $x$ (normalization doesn't have any significant impact on the optimization process) as follows:

$$a_{ij} = \frac{x_{ij}}{||x||_2} \tag{1}$$

Now, following the classical operation, applying the Fourier Transform (FT) along the second dimension ($N_s$), $FT(a_i) = a_i^f$ where $a_i = (a_{ij})_{j \in [1, N_s]}$ and $a_i^f$ denote the corresponding fourier-transformed coefficients. Denoting the maximum number of modes with $K$, intermediate linear transform is a matrix $W \in \mathbb{R}^{N_c \times N_c \times K}$ and $W^k \in \mathbb{R}^{N_c \times N_c}$ denotes this matrix $W$ indexed along the last dimension (corresponding to $k^{th}$ mode). Here, we assume this matrix $W^k$ is orthogonal and thus, can be implemented by a quantum layer. This transformation is now applied to the first $K$ modes along the $N_s$ dimension resulting in the following output:

$$\left[ (W^j a_j^f)_{j \in [1, K]}, (a_j^f)_{j \in [K+1, N_s]} \right] \tag{2}$$

where the vector $a_j^f = (a_{ij}^f)_{i \in [1, N_c]}$. In the classical paper, they discard the rest of modes (make the vectors zero) and we leave them unchanged here. We verified that this doesn't impacts the optimization significantly. For these transformed $K$ vectors $a_j^f \in \mathbb{R}^{N_c}$, we define $b_j^f = W^j a_j^f$ or more explicitly $b_{ij}^f = \sum_t^{N_c} (W_{it}^j a_{tj}^f)$. Given $W^j$, $a_j$ and $b_j$, we can now write the input after Fourier Transform and the intermediate linear transform as:

$$\left[ (b_j^f)_{j \in [1, K]}, (a_j^f)_{j \in [K+1, N_s]} \right] \tag{3}$$

Finally, applying Inverse Fourier Transform ($IFT$) operation on this transformed input, along the $N_s$ dimension, results in the following output as the output of the complete operation:

$$y_i = IFT \left( \left[ (b_{ij}^f)_{j \in [1, K]}, (a_{ij}^f)_{j \in [K+1, N_s]} \right] \right) \tag{4}$$

where $y_i = (y_{ij})_{j \in [1, N_s]}$. The Time Complexity of this complete Fourier Layer ($FT$+linear transform+$IFT$) is $O(N_c + N_s log(N_s))$.

## 3 QUANTUM FOURIER NETWORKS

In this section we propose three quantum circuits to replace the Fourier layer of the FNO, namely the Sequential (Section 3.2.1), Parallel (Section 3.2.2), and the Compound (Section 3.2.3). We compare their computational complexities (see Table 1) and their efficiency in practice in the following sections. Before this, we also define a learnable orthogonal linear transform, using a quantum circuit, called *parameterized butterfly*. Using this as the building block, we propose the Quantum Circuit to carry out the intermediate linear transform proposed in the classical FNO (Section 2).

### 3.1 QUANTUM BUTTERFLY CIRCUITS

An efficient implementation of FFT, proposed by Cooley & Tukey (1965), involves connections in shape of radix-2 butterflies resulting in a logarithmic depth complexity. Taking inspiration from this, we propose such butterfly shaped Quantum Circuits to carry out: a) A learnable linear transform and b) A Quantum Fourier Transform (QFT), both operating only on the unary states. Here each radix-2 operation is implemented using RBS-gates Foxen et al. (2020). Refer supplementary for details regarding the unitary corresponding to this gate.

**Parameterized Quantum Butterfly**. Due to the logarithmic depth offered by the butterfly circuits for FT, which is important given the noisy near-term quantum hardware, we propose such a butterfly shaped Quantum Circuit to carry out a learnable linear transform. Here, each radix-2 butterfly shaped connection is replaced by parameterized RBS gates similar to a concurrent work Cherrat et al. (2022). This results in a orthogonal linear transform only on the unary quantum states similar to the pyramidal circuit proposed by Kerenidis et al. (2021). We further propose a controlled version of these linear quantum layers to perform the operation similar to the intermediate linear transform operation in the fourier layers.

**Unary Quantum Fourier Transform**. Here, we propose a butterfly-shaped Quantum Circuit to carry out the QFT on unary states. It involves replacing each radix-2 operation in the Butterfly FT by a single qubit gate and an RBS-gate with $-\pi/4$ as the angle. Refer supplementary for more details and circuit diagrams for the Parameterized Butterfly layer and Unary QFT.

### 3.2 QUANTUM CIRCUITS FOR FOURIER LAYER

To perform the classical FNO operation using a quantum circuit, the final output state of the circuit should correspond to the quantum-state encoding of the output resulting from this classical operation. Therefore, we load this classical output into a quantum state and for this we use the unary dataloading similar to the one used in Johri et al. (2021) due to its logarithmic depth. It loads an $N$-dimensional classical vector into normalized $N$ quantum states corresponding to the unary representation of numbers from 1 to $N$. The circuit consists of only the Reconfigurable Beam Splitter (RBS) gates. Using controlled version of this circuit, a matrix loader can be defined which we discuss in the supplementary and it loads the normalized vectors comprising the matrix into quantum states. Thus, for the classical output $y = (y_1, \cdots, y_{N_c})$, the encoded quantum state $|y\rangle$ is:

$$|y\rangle = \sum_i |e_i\rangle |y_i\rangle = \sum_i^{N_c} \sum_j^{N_s} IFT\left(\left[(b_{ij}^f)_{j\in[1,K]}, (a_{ij})_{j\in[K+1,N_s]}\right]\right)_j |e_i\rangle |e_j\rangle \tag{5}$$

where $|e_i\rangle$ denotes the state in unary notation with 1 being at the $i^{th}$ position. Note there is no normalization factor, as the inputs $(a_1, \cdots, a_{N_c})$ were assumed normalized in the previous subsection. Now, we discuss the three proposed circuit namely sequential, parallelized and the compound circuit. The sequential circuit replicates the classical operation discussed above and the other two are modifications to it to make the quantum algorithm more efficiently deployable on hardware.

### 3.2.1 SEQUENTIAL QUANTUM CIRCUIT FOR FOURIER LAYER

Figure 1 shows the diagram for this circuit. Lower register of $N_s$ qubits correspond to the second dimension and upper register of $N_c$ qubits correspond to the $N_c$ dimension used in the above mathematical description of the classical operation. We begin by discussing the quantum dataloading followed by quantum transforms corresponding to the classical ones formulated above.
**Dataloading**. Similar to the quantum-state encoding of the classical output defined above, here also

we use the unary matrix loader to load the given input into a quantum state. As shown in Figure 1, the circuit initially contains $N_c$ controlled unary dataloaders, each encoding a row of $N_s$ inputs into the quantum state on the lower register. The loaded state, after controlled operation $X_{N_c}$ in the circuit, can be written as:

$$\sum_i^{N_c} \sum_j^{N_s} a_{ij} |e_i\rangle |e_j\rangle \tag{6}$$

where the coefficients $a_{ij}$ correspond to normalized matrix elements. Denoting the given classical input as $x \in \mathbb{R}^{N_c \times N_s}$ we have (refer supplementary for details):

$$a_{ij} = \frac{x_{ij}}{||x||_2} \tag{7}$$

Now, to apply the $QFT$ operation on lower register, the state can be re-arranged as follows:

$$\sum_i^{N_c} |e_i\rangle \left( \sum_j^{N_s} a_{ij} |e_j\rangle \right) \tag{8}$$

**Quantum Fourier Transform** ($QFT$). Given a normalized real vector $x = (x_i)_{i=1,..M}$, and its Fourier Transform $x^f = (x_i^f)_{i=1,..M}$, the $QFT$ operation in unary basis and its inverse, the $IQFT$ operation, can be defined as follows:

$$QFT(\sum_i x_i |e_i\rangle) = \sum_i x_i^f |e_i\rangle \quad \text{and} \quad IQFT(\sum_i x_i^f |e_i\rangle) = \sum_i x_i |e_i\rangle \tag{9}$$

Applying this $QFT$ operation to the lower register, on the state in eq. 8, we have:

$$\sum_i^{N_c} |e_i\rangle QFT(\sum_j^{N_s} a_{ij} |e_j\rangle) = \sum_i^{N_c} |e_i\rangle \sum_j^{N_s} a_{ij}^f |e_j\rangle = \sum_i^{N_c} \sum_j^{N_s} a_{ij}^f |e_i\rangle |e_j\rangle \tag{10}$$

**Quantum Linear Transform**. To realize the intermediate transform used in the classical operation, we implement an orthogonal matrix $W^k$, corresponding to the one in the classical operation, and realize it using a parameterized butterfly circuit. Furthermore, we propose a "$k$-butterfly" which is a parameterized butterfly circuit ($P_k$) on the top register controlled by the $k^{th}$ qubit of the lower register. As shown in Figure 1, it involves using an ancilla qubit (top-most) initialized to state $|1\rangle$. When this qubit is in state $|1\rangle$, then combining it with upper register results in a hamming weight 2 basis $|h_2\rangle$. Any transformation on this using the $RBS$ gates would result in an another state of the hamming weight 2 basis. We can ignore all these hamming weight 2 states using a post select operation. To apply the $k$-butterfly, we flip this qubit back to 0 if the $k^{th}$ qubit of the lower register is $|1\rangle$, using a CNOT gate. Thus, all the states of upper qubits are in $|h_2\rangle$ basis, except for the ones corresponding to $k^{th}$ unary state of the lower register:

$$\sum_j^{N_s} \sum_i^{N_c} a_{ij}^f |e_i\rangle |e_j\rangle |1\rangle \mapsto \sum_{j=k}^{N_c} \sum_i a_{ij}^f |0\rangle |e_i\rangle |e_j\rangle + \sum_{j\neq k}^{N_c} \sum_i a_{ij}^f |1\rangle |e_i\rangle |e_j\rangle \tag{11}$$

Now, applying $K$ such k-butterfly circuits, we perform the operation $P_j$ on both the unary and the $|h_2\rangle$ states. We consider the unary states in our analysis since we discard the states in other basis:

$$\sum_j^K P_j(\sum_i^{N_c} a_{ij}^f |0\rangle |e_i\rangle) |e_j\rangle + \sum_{j=K+1}^{N_s} (\sum_i^{N_c} a_{ij}^f |1\rangle |e_i\rangle) |e_j\rangle \tag{12}$$

Considering only the top register, the k-butterfly operation $P_j$ corresponds to the sub-matrix $W^j \in \mathbb{R}^{N_c \times N_c}$. The overall matrix $(a_{ij}^f)$ can be decomposed into $N_s$ vectors $a_j^f = (a_{ij}^f)_{i \in [1,N_c]}$. Then, for the first $K$ vectors $a_j^f \in \mathbb{R}^{N_c}$ we will have $b_j^f = W^j a_j^f$, where $b_j^f$ will be the same as in the classical case. Finally, the output state of this circuit after $IQFT$ on the lower register becomes:

$$\sum_i^{N_c} |e_i\rangle IQFT \left( \sum_j^K b_{ij}^f |e_j\rangle + \sum_{j=K+1}^{N_s} a_{ij}^f |e_j\rangle \right) \tag{13}$$

Since $IQFT(\sum_i x_i^f |e_i\rangle) = \sum_i x_i |e_i\rangle$, where $IFT(x^f) = x$, this implies that $j^{th}$ component of $IFT$ would be same as the coefficient of $j^{th}$ state in $IQFT$. From this we can conclude that the state in eq. 5 is equivalent to the state in eq. 13 and thus, this circuit replicates the classical operation. The depth complexity of this circuit is $O((K+1)\log(N_c) + (N_c+2)\log(N_s))$.

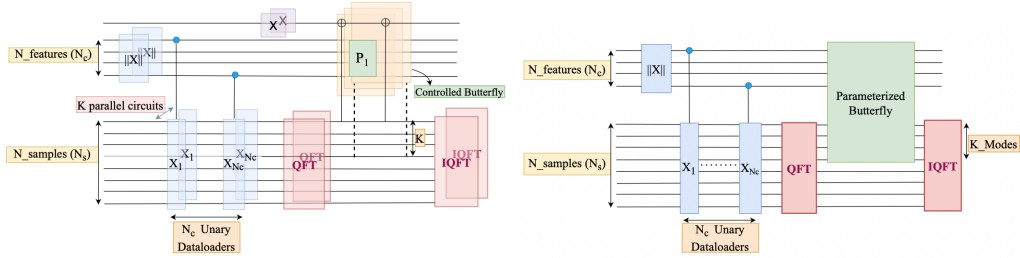

(a) **Left**: Parallelized QFNO Circuit       (b) **Right**: Compound QFNO Circuit

Figure 2: **Left**: Parallelized version of the Sequential Quantum Circuit to minimize the depth of the learning part, thus making it more efficient when deployed on the noisy hardware. For each mode (out of the top $K$) in the transformed input, there is a different circuit to perform the parameterized matrix transform. **Right**: Another variant of the sequential circuit where instead of controlled butterfly circuits, there is a Compound Butterfly Circuit spanning upper register and top $K$ qubits of lower register.

### 3.2.2 PARALLELIZED QUANTUM CIRCUIT FOR FOURIER LAYER

Given the multiplicative noise model for the NISQ devices, the depth of the learnable part, which is proportional to $K$ for the sequential circuit, might be a hindrance to learning and it also increases the computation time. A useful modification then can be parallelizing the learnable butterfly circuits. Figure 2 shows this modified version of the sequential circuit, which consists of $K$ quantum circuits operating parallely and each implementing only one learnable circuit controlled by one of the top $K$ qubits in the lower register. As all the circuits upto the learnable part are similar to the sequential circuit, we can directly write the state after the QFT, using eq. 10, as:

$$\left[ \sum_j^{N_s} \sum_i^{N_c} (a_{ij}^f)_k \, |e_i\rangle_k \, |e_j\rangle_k \right]^K_{k=1} \tag{14}$$

where the index $k$ denotes the $k^{th}$ parallel circuit. Also, in the $k^{th}$ parallel circuit, the learnable butterfly part is controlled by the $k^{th}$ qubit of the lower register. We recall that the pyramid applied on the top register is effectively mapping the vector $a_j^f$ to $b_j^f$ (see Eq.13) and thus we can write the updated state of the circuits as:

$$\left[ \sum_{j \neq k} \sum_i^{N_c} (a_{ij}^f)_k \, |e_i\rangle_k \, |e_j\rangle_k + \sum_i^{N_c} (b_{ij}^f)_k \, |e_i\rangle_k \, |e_k\rangle_k \right]^K_{k=1} \tag{15}$$

Now, applying IQFT on the lower register in each of the circuits independently:

$$= \left[ \sum_i^{N_c} |e_i\rangle_k \, IQFT \left( (b_{ik}^f)_k \, |e_k\rangle_k + \sum_{j \neq k} (a_{ij}^f)_k \, |e_j\rangle_k \right) \right]^K_{k=1} \tag{16}$$

In supplementary, we derive that this joint output of these circuits cannot lead to the output of the classical operation even after measurement and classical post-processing. We also show that how measuring without applying the $IQFT$ operation and instead applying the classical $IFT$, after some post-processing, can lead to the same output as the classical operation. The depth of this parallel version of sequential circuit is $(2)\log(N_c) + (N_c + 2)\log(N_s)$ and a total of $K$ quantum circuits are required to execute this parallelly.

### 3.2.3 COMPOUND QUANTUM CIRCUIT FOR FOURIER LAYER

As highlighted in previous subsection, depth of the parameterized part of the sequential circuit might make the learning process difficult on currently available noisy quantum hardware. To deal with

this, we propose another variant of the sequential circuit, where, instead of having the parameterized circuit controlled by top $K$ qubits of lower register, we span a a single bigger pyramid over the upper register ($N_s$) and top $K$ qubits in the lower register. Note the upper and lower registers are unary independently. Therefore, if we jointly consider the upper register and top $K$ from the lower register, then there would be states with hamming weight two as well, corresponding to the scenario when 1 is in one of the top $K$ qubits of the lower register. Thus, instead of a unary basis (which corresponds to having hamming weight only one), we have hamming weight one and hamming weight two basis. Lets imagine a butterfly circuit on $N_c + K$ qubits. The complete unitary will be a $2^{N_c+K} \times 2^{N_c+K}$ block diagonal matrix with each block corresponding to a subspace with fixed hamming weight Kerenidis & Prakash (2022), $B = B_1 \otimes B_2 \otimes ... \otimes B_n$, where $B_i$ correspond to the block diagonal unitary for subspace with hamming weight $i$. Since our input has hamming weight 1 or 2, we only care about unitaries $B_1$ and $B_2$. $B_1$ will be of size $(N_c+K) \times (N_c+K)$ and $B_2$ of $\binom{N_c+K}{2} \times \binom{N_c+K}{2}$. Given these unitaries, we take the input state and split it into $(N_c + K)$ hamming weight 1 states and $\binom{N_c+K}{2}$ hamming weight 2 states. Then we apply the $B_1$ and $B_2$ to these states. Also, $B_1$ is a butterfly matrix and $B_2$ is the corresponding compound order two matrix Horn & Johnson (2012). Figure 2 shows the diagram for this circuit. Given the circuit is similar to the sequential circuit till the $QFT$ operation, the state of this circuit would be same as the one in eq. 10. We now separate this complete state into two sets of states corresponding to hamming weight 1 and 2:

$$= \sum_i^{N_c} \sum_j^K a_{ij}^f |e_i\rangle |e_j\rangle + \sum_i^{N_c} \sum_{j=K+1}^{N_s} a_{ij}^f |e_i\rangle |e_j\rangle \tag{17}$$

where the first term corresponds to hamming weight 2 states $|h_2\rangle$ and similarly the second term corresponds to hamming weight 1 states $|h_1\rangle$. Lets first focus on the term corresponding to $|h_1\rangle$. It does not contain the states where the qubits in upper register are all 0 and the 1 lies in the top $K$ qubits of the lower register. It implies that the coefficients of all these states should be taken as zero. Therefore, the state corresponding to this $|h_1\rangle$ can also be written as:

$$\sum_i^{N_c} \sum_{j=K+1}^{N_s} a_{ij}^f |e_i\rangle |e_j\rangle + \sum_i^K \sum_{j=K+1}^{N_s} 0 |e_0\rangle |e_{ij}\rangle \tag{18}$$

where $|e_0\rangle$ denotes the state corresponding to no ones in the upper $N_c$ register and $|e_{ij}\rangle$ denotes the hamming weight 2 state for the lower register, where $i$ and $j$ denote the positions of 1. Similarly, if we consider the first term in eq. 17 corresponding to $|h_2\rangle$, we further have to include states where both ones are in upper register or both ones in top $K$ of the lower register. These new states again would have zero coefficients. As a result, we can write the term corresponding to $|h2\rangle$ in eq. 17 as:

$$\sum_i^{N_c} \sum_{j>i}^{N_c} 0 |e_{ij}\rangle |e_0\rangle + \sum_i^{N_c} \sum_j^K a_{ij}^f |e_i\rangle |e_j\rangle + \sum_i^K \sum_{j>i}^K 0 |e_0\rangle |e_{ij}\rangle \tag{19}$$

This results in a total of $\binom{N_c+K}{2}$ states. Now, we apply the butterfly circuit, corresponding to unitary $B_1$, to the $|h_1\rangle$ state in eq. 18. For notational consistency we denote this operation as a multiplication with matrix $W^1 \in \mathbb{R}^{(N_c+K) \times (N_c+K)}$. It results in the transformed coefficients $b_{ij}$:

$$b_{ij}^f = \sum_t^{N_c} (W_{it}^1 a_{tj}^f) + \sum_{t=N_c+1}^{N_c+K} (W_{it}^1 \times 0) \; i \in [1, N_c + K] \; j \in [K + 1, N_s] \tag{20}$$

Furthermore, we also apply a post select operation to preserve the basis, selecting only the states with non-zero coefficient before applying the $B_1$. Now, for the hamming weight 2 state, the second order compound unitary $B_2$, denoted by the matrix $W^2 \in \mathbb{R}^{q \times q}$ where $q = \binom{N_c+K}{2}$, has each of its elements corresponding to the determinant of a $2 \times 2$ submatrix of $W^1$:

$$W_{i,j}^2 = W_{a,b}^2 W_{a+k,b+k}^2 - W_{a+k,b}^2 W_{a,b+k}^1 \tag{21}$$

for some $a, b, k < N_c + K$. After applying this unitary $B_2$ on $|h_2\rangle$ states, their coefficients $c_{ij}$ are:

$$c_{ij} = \sum_t^{\binom{N_c}{2}} (W_{it}^2 \times 0) + \sum_t^{N_c K} \left( W_{i,t+\binom{N_c}{2}}^2 \times a_{tj}^f \right) + \sum_{t=N_c K+\binom{N_c}{2}}^{\binom{N_c+K}{2}} (W_{it}^2 \times 0) \tag{22}$$

Similar to the case of hamming weight 1, here also we use a post select operation to discard the states which initially had coefficients zero thereby preserving the basis. Combining the transformed $|h_1\rangle$, $|h_2\rangle$ states and applying $IQFT$ on the lower register, the final output state of this circuit is:

$$\sum_i^{N_c} |e_i\rangle \, IQFT \left( \sum_j^K c_{ij} |e_j\rangle + \sum_{j=K+1}^{N_s} b_{ij} |e_j\rangle \right) \tag{23}$$

and the order of the depth complexity is $\log(N_c + K) + \log(N_c) + (N_c + 2)\log(N_s)$.

## 4 EXPERIMENTS

In this section, we analyze our proposed Quantum algorithms for solving PDEs and Image classification tasks. We compare them against the *state-of-the-arts* in both the domains, *i.e.*, classical Fourier Networks (PDEs) and CNNs (image classification), for both the tasks. All the details related to architecture/hyperparamaters are provided at the end of the paper. All the experiments shown in this section are simulated, i.e., the quantum operations have been simulated using classical matrices corresponding to quantum unitaries. The currently available quantum hardware is limited to 8-10 qubits and is also too noisy for deeper circuits. Deploying these algorithms on a quantum hardware will involve noise which can be either due to noisy quantum gates, measurement, environment, etc.

### 4.1 PARTIAL DIFFERENTIAL EQUATIONS

We show results on all the three PDEs used in the classical Fourier Layer paper Li et al. (2020): Burger's equation, Darcy's Flow equation and Navier-Stokes equation, using the datasets proposed in that paper. All of these equations were designed for modelling the flow of fluids and have found their applications in other domains as well. We abstractly describe the three tasks. For equations and other details please refer Li et al. (2020). We analyze the performance of the trained networks across different resolutions ($N_s$) for the first two and for different viscosity values for the third.
**Burger's Equation**. It is a 1D-Partial Differential Equation for modeling fluid motion. The initial condition of the fluid is represented by a function. We need to learn the mapping from this function to the solution function at time one, for a given viscosity. Figure 3 (left) shows the comparison

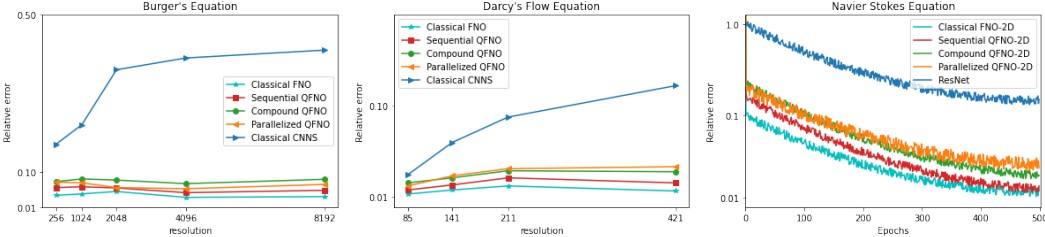

Figure 3: **Left**: Performance comparison (relative error as used in Li et al. (2020)) of the classical fourier networks, CNNs and the three circuit proposals for a quantum fourier layer on the Burger's 1D PDE equation across different resolutions. The quantum fourier circuits are quite close to the performance of the classical fourier baseline and much better than classical CNNs in minimizing the error. **Middle**: Same comparison on the Darcy's 2D PDE for different resolutions. A similar relative performance is observed where error in CNNs is much larger and is increasing w.r.t. resolution, whereas the error in the other four (3 quantum circuits and classical layer) is quite similar. **Right:** Convergence comparison for the Navier-Stokes equation with $v = 1e - 3$, trained for $500$ epochs.

of relative error in estimating this mapping among the classical Fourier Layer, classical CNNs and proposed quantum circuits for the Fourier Layers, across different resolutions. The quantum circuits perform comparably to the classical Fourier Layer and much better than classical CNNs.
**Darcy's Flow Equation**. For this case, it is a 2D PDE where the aim is to learn the mapping from the diffusion coefficient function to the solution function in presence of a forcing function. All of them are functions of positional coordinates only. Figure 3 shows the relative error for the 2D-version of all the methods in solving this PDE, across different resolutions. Here also the three quantum

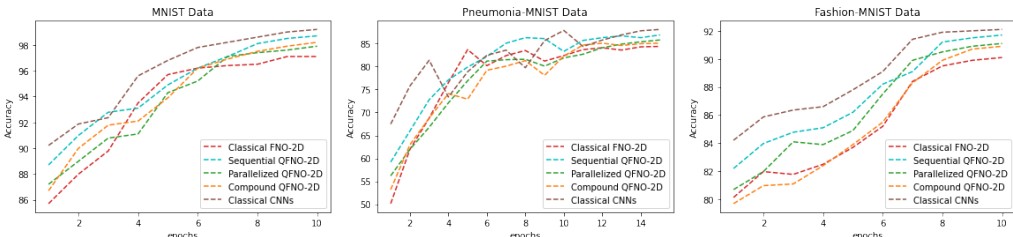

Figure 4: **Left**: Performance comparison of the CNNS, classical fourier layer and the proposed quantum circuits on the MNIST dataset. It can be observed that all of them perform quite similarly, classical CNNs being the best. **Middle**: A similar comparison on the Pneumonia-MNIST Yang et al. (2021) dataset. The performance of CNNs is somewhat noisy here whereas it is smoother in case of the sequential circuit, both converging to a similar value. The compound quantum circuit and the classical Fourier baseline are also quite close to the CNNs in convergence. **Right**: Same comparison on the FashionMNIST Xiao et al. (2017) data. Here, a significant difference in the performance is observed with CNNS being the best followed by the Sequential circuit.

circuits and the classical fourier layer show similar performance, consistent across resolutions, and the CNNs show much worse results, having error increased with resolution.

**Navier Stokes Equation**. Similar to Li et al. (2020), we consider the 2D-navier Stokes Equation where the aim is to model the fluid vorticity upto instant $T(> 10)$ given the vorticity upto time 10. Figure 3 (right) shows the performance comparison for this equation between our proposed circuits and classical methods. It shows the convergence comparison for this family with viscosity $\nu$ fixed to $1e - 3$ for all the methods. Here again, it can be observed that all the proposed circuits and the classical Fourier method perform significantly better than CNNs. Also, from Table 2, sequential circuit performs similarly to the classical method and the others converge at a slightly higher error.

Table 2: Comparison of parameters required by one layer of the proposed circuits and the existing classical Fourier Layer along with error analysis for different $\nu$ and $T$ values for the 2D case of a Navier Stokes equation.

| Method | Classical FNO | Sequential QFNO | Parallelized QFNO | Compound QFNO |
|---|---|---|---|---|
| Parameters | 294,912 | 23,040 | 23,040 | 6,144 |
| $\nu = 1e - 3; T = 50$ | 0.0139 | 0.0148 | 0.0167 | 0.0186 |
| $\nu = 1e - 4; T = 30$ | 0.1603 | 0.1618 | 0.1633 | 0.1660 |
| $\nu = 1e - 5; T = 20$ | 0.1601 | 0.1615 | 0.1626 | 0.1638 |

## 4.2 IMAGE CLASSIFICATION

We further compare our proposed Quantum algorithms on the downstream image classification tasks, as a side experiment, on benchmark datasets including the MNIST , FashionMNIST Xiao et al. (2017) and PneumoniaMNIST Yang et al. (2021) datasets. Figure 4 shows the results for this evaluation. It can be observed that our proposed algorithms perform decently in the image-classification task as well.

## 5 CONCLUSION

We proposed a quantum algorithm to carry out the recently proposed classical Fourier Neural Operator on the quantum hardware. We further proposed two more quantum algorithms, which perform a different operation than the classical algorithm and can be much more efficiently deployed on a quantum hardware. The aim is to make the learning process more efficient when using the noisy quantum hardware. Experimental results further verify that proposed quantum circuits perform efficiently when solving PDEs. The sequential circuit is quite similar to the best performing classical algorithm for PDEs and performs decently on image classification as well. An interesting future direction can be further modifying the learning process of the compound circuit so that it outperforms the sequential circuit at the same time being more efficient to deploy.

## REPRODUCIBILITY STATEMENT

We have used the JAX framework to develop the methods discussed in the paper. All the experiments shown in the paper are simulated by defining the matrices using the unitaries corresponding to the quantum gates used in the paper. The architectural and hyperparameter details for both classical and quantum Fourier Layers are same as the ones used in Li et al. (2020). All the experiments involve a 4-layered architecture corresponding to both classical Fourier baseline and quantum Fourier circuits. The architecture for CNNs used in the Burger and Darcy Flow PDE experiement is a Fully Convolution Networks based architecture proposed in Zhu & Zabaras (2018). For the Navier stokes, we use the ResNet architecture for CNNs, comprising 18 Residual Blocks as used in Li et al. (2020). For the CNNs in the image classification tasks, we use a four layered architecture with 64 channels and also use a max-pooling layer after the second and fourth layer. For both the tasks, each classical or quantum layer is immediately followed by a ReLU activation and Batch Normalization operation. For the classification tasks, finally the output of each of the architectures is fed to a linear layer with projection heads equal to number of classes for that task. For the classical FNOs and proposed quantum FNOs except the compound quantum FNO, $K$ is fixed to 16 for 1D PDE and image classification and 12 for 2D PDEs, and $N_c = 64$ for the 1D PDE and 32 for the 2D PDEs. For image classification, $N_c$ is set to 16. For the compound quantum FNO, due to less parameteric requirements $K$ is set to 16 and $N_c$ to 48 for all the tasks. The input for the PDEs having $N_c$ initially 1 is first projected to the the required $N_c$ using a linear layer for all the classical and quantum fourier methods similar to Li et al. (2020). The final output of all the layers of $N_c$ features is back projected to $N_c = 1$ using another linear layer. For all the PDEs, unless specified, 1000 train instances and 200 test instances are used and training is carried out for 500 epochs using Adam optimizer with learning rate initialized to 0.001 and halved after every 100 epochs. It is fixed to 0.001 for the classification task. Number of epochs for the classification tasks can be seen in the plots.

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

# Appendix

## A    QUANTUM BUTTERFLY CIRCUITS

As discussed in the paper, replacing each radix-2 butterfly shaped operation in the butterfly diagram for Fast Fourier Transform (FFT) by a parameterized 2-qubit gate, Reconfigurable Beam Splitter $(RBS)$ gate, results in a parameterized butterfly circuit (See Figure 5). The gate is parameterized by a single parameter $\theta$ and the corresponding unitary is:

$$RBS(\theta) = \begin{pmatrix} 1 & 0 & 0 & 0 \\ 0 & cos\,\theta & sin\,\theta & 0 \\ 0 & -sin\,\theta & cos\,\theta & 0 \\ 0 & 0 & 0 & 1 \end{pmatrix} \tag{24}$$

It can be observed that it applies transformations on the $|01\rangle$ and $|10\rangle$ basis and is an identity operation on the remaining. These both comprise the unary basis or hamming weight 1 basis for two qubits. It can then be understood that it applies transformation between the states with same hamming weight basis and thus, preserves the hamming weight. Therefore, if we load our input to a basis with a certain hamming weight, then after any transformation made of these $RBS$ gates, the basis states corresponding to non-zero coefficients in the output will have that same hamming weight. Corresponding to $N$ qubits would be only $N$ unary states and thus, if an $N$-dimensional input is encoded into $N$-unary states using the unary loading similar to Johri et al. (2021), then the transformation resulting from this butterfly circuit can be seen as a multiplication of the coefficients of these $N$ unary states with an orthogonal matrix $W \in \mathbb{R}^{N \times N}$. A recent work by Kerenidis et al. (2021) further explains this orthogonal matrix multiplication for a pyramid-shaped circuit.

**Unary QFT**. As shown in Figure 5 (Right), replacing the $RBS$ gate in parameterized butterfly circuit with a fixed single qubit gate and a fixed angle $RBS$ gate results in a circuit which can perform $QFT$ in the unary basis. The depth complexity of this circuit is still O($\log(M)$) where $M$ is the number of qubits and also the total number of unary states. The single qubit gate is just a Phase gate which rotates about the $Z$-axis with angle set to $-\frac{2\pi k}{N}$ and matrix corresponding to this gate is: $\begin{pmatrix} 1 & 0 \\ 0 & e^{-i\frac{2\pi k}{N}} \end{pmatrix}$ where $N$ and $k$ corresponding to each $\omega$ are same as in the corresponding radix-2 operation in the classical butterfly Fourier Transform ($\omega_N^k$).

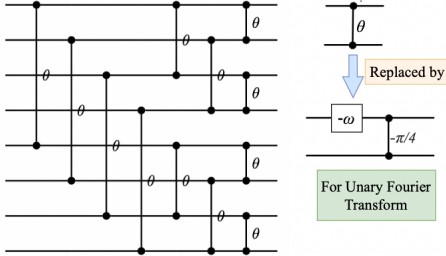

Figure 5: **Left**: Parameterized Quantum Circuit generated by replacing each connection in the Butterfly Diagram for FFT by 2-qubit $RBS(\theta)$ gate, where $\theta$ is a learnable parameter unique to each gate. As discussed in the text, it acts as an orthogonal feed-forward neural network layer which transforms the coefficients of unary basis states. **Right:** Replacing the 2-qubit parameterized RBS gate by a single qubit rotation gate and RBS gate, with angle fixed to $-\pi/4$, results in the circuit for doing QFT on the unary states.

## B    DATALOADERS

Since our input is a $N_c \times N_s$ matrix, to load it into a quantum state, the matrix can be interpreted as $N_c$ vectors each of size $N_s$. The resultant should be a superposition of these vectors. In the paper we discussed a recent work Johri et al. (2021) which used dataloaders to load $N$ dimensional input

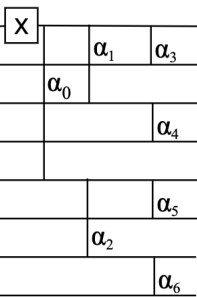

Figure 6: Quantum circuit for loading a classical vector into unary basis Johri et al. (2021). It comprises of $(N-1)$ $RBS$ gates where the angles for each of these gates ($\alpha_i$) are pre-calculated using the input. See Johri et al. (2021) for further details.

to $N$ unary quantum states. One of the proposed dataloaders is shown in Figure 6 for 8 qubits. It involves firstly flipping the first qubit to state $|1\rangle$. Then it involves applying (N-1) $RBS$ gates with precomputed angles ($\alpha_i$) for loading this vector. It can be observed that it is a $log$-depth circuit. For a classical input vector a = $(a_1,...,a_N)$, it results in the following quantum state:

$$|a\rangle = \frac{1}{||a||} \sum_i^N a_i |e_i\rangle \tag{25}$$

where $|e_i\rangle$ denotes the unary basis states for $N$ qubits. There will be total $N$ such state and coefficient of each of these states would correspond to the normalized classical vector element. To load the input matrix $b \in \mathbb{R}^{N_c \times N_s}$ we can superpose its row vectors $b_i$, where each of these vectors would be first loaded using some controlled vector-loader. To do so, we first load the norms corresponding to each of these rows $||b_i||$ using the norm-loader shown in the circuit diagrams in the paper. It results in the following state on the upper register:

$$\frac{1}{||b||} \sum_i^{N_c} ||b_i|| |e_i\rangle \tag{26}$$

Now, for each of the unary states $|e_i\rangle$ in the upper register, we load the corresponding row vectors $(b_{ij})_{j \in [1,N_s]}$ in the lower register using a controlled version of the unary vector loader (controlled operations labelled $||X_i||$ in the circuit diagrams) which involves applying the unary vector loader and its adjoint along with CNOT gates. For more details on this loader, refer Cherrat et al. (2022). The final loaded state after $N_c$ of these controlled unary vector loaders is:

$$\frac{1}{||b||} \sum_i^{N_c} \sum_j^{N_s} b_{ij} |e_i\rangle |e_j\rangle \tag{27}$$

## C  PARALLEL QUANTUM CIRCUIT

Given the final state, after the IQFT, for the parallelized circuit in Eq.16 in the paper:

$$= \left[ \sum_i^{N_c} |e_i\rangle_k IQFT \left( (b_{ik}^f)_k |e_k\rangle_k + \sum_{j \neq k} (a_{ij}^f)_k |e_j\rangle_k \right) \right]_{k=1}^{K} \tag{28}$$

We denote coefficients of this state corresponding to $k^{th}$ circuit by $(c_{ij})_k$ and thus re-writing it as:

$$= \left[ \sum_i^{N_c} \sum_j^{N_s} (c_{ij})_k |e_i\rangle_k |e_j\rangle_k \right]_{k=1}^{K} \tag{29}$$

where $(c_{ij})_k$ are explicitly given by using the equation for the fourier transform as follows:

$$(c_{ij})_k = \frac{1}{N_s} \left( \sum_{j \neq k} (a^f_{ij})_k e^{i\frac{2\pi j}{N_s}} + (b^f_{ik})_k e^{i\frac{2\pi k}{N_s}} \right) \tag{30}$$

Similarly writing $c_{ij}$ for the sequential circuit discussed in the paper (using eq. 13):

$$c_{ij} = \frac{1}{N_s} \left( \sum_{j=K+1}^{N_s} a^f_{ij} e^{i\frac{2\pi j}{N_s}} + \sum_{j}^{K} b^f_{ij} e^{i\frac{2\pi j}{N_s}} \right) \tag{31}$$

Comparing the above two equations leads to the observation that coefficients in $Eq$. 31 wouldn't be a subset of coefficients in Eq. 30 and there is no closed form classical processing/transformation to achieve this. Thus, this parallel circuit results in a somewhat different operation which is intuitively similar to the sequential circuit. Given, the experimental results, this operation is also effective in dealing with PDEs/Images, at the same time being more effective than sequential circuit under noisy scenarios. However, if we remove the $IQFT$ operation from this circuit and instead apply the classical $IFT$, measuring after eq. 15, we get the following K matrices after applying the square root operation:

$$\left[ (b^f_k)_k, (a^f_j)_{j \neq k} \right]_{k=1}^{K} \tag{32}$$

where $b^f_j$ and $a^f_j$ have been defined previously. There are $K$ such $N_c \times N_s$ matrices. In case we combine $b^f_k$ from all of the $K$ matrices with $(a^f_j)_{j \in [K+1, N_s]}$ from any of the $K$ matrices, suppose the first one, it leads to the following $N_c \times N_s$ matrix:

$$\left[ (b^f_j)_{j \in [1,K]}, (a^f_j)_{j \in [K+1, N_s]} \right] \tag{33}$$

which is exactly same as eq. 3. Thus, this circuit (without the IQFT) followed by some classical post-processing and IFT can replicate the classical fourier layer operation.

## D   WHY COMPOUND MATRIX ?

The intuition behind why a different operation than classical, in form of this compound matrix, might work lies in the expressive power of this operation. The matrix for the sequential circuit can be interpreted as a special case of this compound matrix where instead of trainable connections, upper and lower registers interact using a fixed control gate. Due to this special connection, this transform is making the convergence efficient. This compound matrix, thus, is a more expressive version of the linear transform and also with a lower depth. Using it was further backed by the assumption that if optimized carefully it might lead to either a more parallelized version of the sequential circuit or a completely different and a more efficient learned matrix. Also, this is more a quantum-native operation which would be quite complex to carry out classically and thus can provide a better demonstration of quantum-advantage.

