# OpenReview forum: "Quantum Fourier Networks for solving Parametric PDEs"
_ICLR.cc/2023/Conference — Submitted to ICLR 2023_

### Official Review · Reviewer_EV9o · 2022-10-22

**Confidence:** 3
**Correctness:** 3
**Technical Novelty And Significance:** 3
**Empirical Novelty And Significance:** 3
**Recommendation:** 6

**Clarity, Quality, Novelty And Reproducibility:**

The paper is interesting and some implementation details are given, while the authors do not show or plan to release any source code which I personally think is an important issue in existing quantum machine learning literature (not only this paper). I strongly suggest the authors could release their code otherwise the impact of the work can be less prominant.

For Section 4.2, I am not quite sure how it is implemented for image classifition.

**Strength And Weaknesses:**

Strength:
1) good topic and quamtum AI solver for PDEs is promising and worh study
2) complete evaluation on three typical kinds of PDEs with additional test on image classification
3) the technical approach seems convincing with good justification

Weakness:
1) the AI PDE solvers e.g. ONet and FNO are all open-sourced, this work may also release the code for advancing the research in this intersection area?
2) some details might be missing in the paper, e.g. the image classification implementation part.
3) it still gives me an impression as a combiation of existing techniques i.e. FNO and quantum Foruier transformation

Overall, I think this is a good paper, and I vote for acceptance.

**Summary Of The Paper:**

This paper proposes three tailored quantum circuits, inspired by the FNO (Fourier Neural Operator), to learn the functional mapping for PDEs (Partial Differential Equations). The authors evaluate the proposed methods on three PDE families, with results showing that the quantum methods are comparable in performance to the classical FNO. This method is further verified on image classification task with comparble performance to CNN, which shows the potential value of their method to other domains.

**Summary Of The Review:**

The paper addresses an interesting and promising area for adapting quantum computing to solve the machine learing-based PDE solvers. Such AI solvers themselves have attracted intensive attention in both computer science and applied math communities and derive two main lines of research: ONet and FNO. This paper presents a quantum version of FNO based the natural quantum-friendly computing of Fourier transformation and its inverse. The experiments over three typical PDEs and it can also be applied in other domains as preliminarily verified on the image classification task. The experiments are all done on simulation based classic computers for the unavailability to high-end quatum computers (with challenges in terms of noise and qubit limites etc.), which I think is understandable.

---

> ### Author Response · Authors · 2022-11-16
> **Reply to reviewer's concerns**
>
> We thank the reviewer for the constructive review. Below we provide answers to some of the raised concerns,
>
> 1. Open Sourcing Code for advancement in research
>
> Thank you for raising this concern. We will be very happy to open source this work to promote research in quantum AI solvers for PDEs and already have planned to release a well documented code for this paper upon publication.
>
> 2. Image Classification Implementation
>
> The experimental and architectural details for this are given at the end of paper. We apologise in case it was not clear from this section and other details given in the paper. We simulate the matrices required in 2D version of classical FNO (please refer the open sourced implementation) using quantum circuits. These matrices expect a 2D input and thus, for the classification task, instead of a 1-channel map denoting initial condition/coefficients as 2D locations, it is now an image corresponding to the used datasets. Following the classical FNO, we just extrapolate the 1D Quantum Fourier transform described in the paper to 2D and similarly the intermediate linear transform is also extended to 2D, where each parameterised butterfly circuit still implements a different feature transformation layer but now for a given combination of mode along the two spatial dimensions. Therefore we now have KxK such independent parameterised circuits.

---

### Official Review · Reviewer_ExUz · 2022-10-25

**Confidence:** 2
**Correctness:** 2
**Technical Novelty And Significance:** 2
**Empirical Novelty And Significance:** 2
**Recommendation:** 3

**Clarity, Quality, Novelty And Reproducibility:**

- Complexity bound for the classical FNO. Page 3 : "The Time Complexity of this complete Fourier Layer (F T+linear
transform+IF T) is O(K + 2Nslog(Ns))." There are $K$ many matrices multiplication with size $N_c \times N_c$, the complexity should be $O(KN_c^2)$. Am I missing something? When using the big-O notation, there is no need to put a constant inside the `O', namely,  the `2` in front of $N_s\log (N_s)$

- Presentation in Sec 2 (the classical FNO) is confusing and seems to contain errors.
1. Notational-wise. I never see using superscript $f$ to represent Fourier transform, which is quite non-standard. The use of subindices $i, j$ coupled with $f$ makes it very hard to parse the notation. In addition, the same letters $i, j$ seem to be used in both the original and Fourier domain.
2. Equations (2) and (3) seem wrong to me, compared to equation (5) in the FNO paper (https://arxiv.org/pdf/2010.08895.pdf). It reads the authors take the first $K$ data points of the first transform rather than the first $K$ modes. Isn't there should be a $f$ for all $a_j$ for $j>K$ in equations (2) and (3)?

- Empirically, how can I tell if the proposed method is *substantially faster* than the classical counterpart. I did not see empirical support.

- In all PDEs experiments, the classical FNO is better than the quantum counterpart.

- What are the scales (linear vs. log) used in the $y$-axis in Fig 3? The Burger plot uses a `linear` scale, while the Navier stoke's equaiton uses a `log` scale. I can't tell what scale is used for Darcy's flow.

- In addition, I really think the `blue curve` (CNN architecture) should be removed from the plot so that we could see the high resolution of the (quantum) FNO. The error from CNN is one order of magnitude larger than the others, which makes it hard to see the finer difference between the proposed and the baseline methods.

- The experimental results in image classification are unconvincing for many reasons below. In particular, "Experimental results further verify that proposed quantum circuits perform efficiently in both solving PDEs and image classification" is an overstatement.
1. I did not see papers using FNO for image classification. If so, is there a common baseline? If not, how can I trust the models in the experiments are well-tuned and near-optimal?
2. Using MNIST (and siblings) for image classification is unconvincing due to its simplicity. A more complicated dataset, e.g., CIFAR10/100, should be used.
3. I am not sure if the result from the proposed method is comparable to CNNs. Different CNN architectures will give a very different performance, which also depends on many other factors (e.g., data augmentation, batch-normalization, learning rate, batch size, etc.) To make a claim like "It can be observed that our proposed algorithms are better than the classical Fourier Layer and comparable to
CNNs, especially the sequential circuit, thereby proving to be effective in the vision domain as well," requires **rigorous** and **comprehensive** empirical study.





**Strength And Weaknesses:**

- Presentation in Sec 2 (the classical FNO) is confusing, both notational-wisely and mathematically.
- The experimental results are not convincing enough to support many claims in the paper.
- The complexity bound, at least for the classical setting, seems not correct

Please see the details below.

**Summary Of The Paper:**

Fourier Neural Operator (Li 2020) has become a popular tool in solving PDEs numerically. The main contribution of the paper is to replace the classic Fourier Layer in FNO with the quantum Fourier layer. The authors argue that the proposed algorithm is provably substantially faster (using quantum hardware) than the classic counterpart. The authors also provide several experimental results (solving 3 PDEs and one simple image classification) to show that the proposed method is comparable to the classical FNO.

I am not familiar with quantum computing and won't be able to judge the significance of the theoretical contribution. Nevertheless, the overall quality of the paper seems not to be very high, which will be detailed below. Moreover, the experimental results from the paper do not support the strong claims made by the authors.

**Summary Of The Review:**

Overall, the quality of the paper seems not to be high and I have raised several concerns.

---

> ### Author Response · Authors · 2022-11-16
> **Rebuttal Revision**
>
> Dear reviewer, please see the revised copy of the paper with equations 2 and 3 corrected. Also, the time-complexity for the classical part had a small mistake which has been updated. Finally, given the concerns, we have rephrased the claims for image classification in section 1 (last paragraph), section 4.2 and section 5

---

> ### Author Response · Authors · 2022-11-16
> **Reply to reviewer's comments and concerns**
>
> We thank the reviewer for their review. Below we provide answers to raised concerns.
>
> 1. Complexity bound for classical algorithm.
>
> Thank you for raising this concern. The time complexity of the classical part had a small error. It should be O(*$N_c$* + 2*$N_s$*log($N_s$)). The complexity of the intermediate linear transformation is replaced by O($N_c$) instead of $K$. This can be accounted to the fact that matrix multiplication operation, corresponding to each of the $K$ matrices can be parallelized and this will reduce the complexity of multiplying the $K$ $N_c\times N_c$ matrices with $N_c$ sized vectors to be O($N_c$).
>
> 2. Equation 2 and 3 seems wrong compared to equation 5 in the FNO paper.
>
> We thank the reviewer for pointing it out, it is indeed a mistake. The transform is applied to all the samples which we have mentioned in the text. Also our equations in the quantum part are also derived keeping this in mind. We have corrected it in the revised version.
>
> 3. Empirically, how can I tell if the proposed method is substantially faster than the classical counterpart. I did not see empirical support.
>
> The numerical experiments presented in this work were performed on a classical simulator of the quantum hardware. Therefore, it is not possible to witness the theoretical speedup proven in the paper. Even with quantum hardware, it would have been hard to see it, as the current devices are noisy and slow (clock rate wise) compared to modern GPUs. This work, as any in the quantum computing field, is waiting for longer term development to show practical benefit in terms of speedup. We show the potential of quantum computers to offer faster or better tools for solving real world problems upon their advent.
>
> 4. What are the scales (linear vs. log) used in the $y$-axis in Fig 3?
>
> Thanks for the concern. The scale for the Darcy's flow equation along the y-axis is same as that for Navier Stokes equation, i.e. log scale.
>
> 5. The experimental results in image classification are unconvincing for many reasons below. In particular, "Experimental results further verify that proposed quantum circuits perform efficiently in both solving PDEs and image classification" is an overstatement.
>
> The reviewer’s concern is shared by the authors, and some claims have been revised accordingly. Regarding the use of (Q)FNO for image classification: we wish to have more image classification examples in the future (ImageNet, CIFAR-10). We have provided the MNIST example as a simple baseline for future work. In addition, we want to emphasize that this work targets solving PDEs primarily, whereas image classification is just a side, interesting, application. Some claims have been rephrased in the paper to make this more explicit. Moreover, we would also like to mention that we have not tried to find the best model parameters for these applications and have worked mostly on providing a theoretical model for QFNOs.
>
> 6. In all PDEs experiments, the classical FNO is better than the quantum counterpart.
>
> The motivation of this paper is to develop theory using the tools and constraints provided by quantum computing, for a possible quantum counterpart of the classical FNO. It doesn’t claim to be always better in terms of performance but instead, an algorithm similar in performance to the current classical state-of-the-art, which will be much faster for large scale scenarios upon availability of real quantum hardware.

---

> > ### Comment · Reviewer_ExUz · 2022-11-22
> > **reply**
> >
> > Thanks a lot for several clarifications.
> >
> > One minor comment: it might be easier for reader to parse a plot if all subplots are in the scale.

---

### Official Review · Reviewer_URvi · 2022-10-26

**Confidence:** 3
**Clarity, Quality, Novelty And Reproducibility:** I find the clarity of the paper can b…
**Correctness:** 4
**Technical Novelty And Significance:** 3
**Empirical Novelty And Significance:** 2
**Recommendation:** 5

**Strength And Weaknesses:**

While it is interesting to see connections between quantum algorithms and classical PDEs, I am not sure if this extension is well-motivated. Empirically, in classical FNO, the FFT is already quite fast and the matrix multiplication takes the most computation since the channel dimension of FNO can be very large. It's not very sure if quantum FNO can be more efficient in practice.

Strength:
- nice observation to speed up FFT with quantum algorithms.
- empirically, the Quantum FNO has comparable performance, with a small number of parameters.

Weakness:
- The Quantum FFT seems not to be justified. Unnecessary to use quantum language when everything is classical.
- empirically, the performance is not as good as the original FNO.
- the Mnist example looks unnecessary. There exists a previous work on Imagenet (https://arxiv.org/pdf/2111.13587.pdf).
- there exists a previous work that uses matrix butterfly decomposition for neural operators (https://proceedings.mlr.press/v162/dao22a.html).


**Summary Of The Paper:**

In this paper, the authors extend the classical Fourier neural operator with Quantum circuits. It applies butterfly circuits as a model compression. The paper studies Burgers, Darcy, and Navier-Stokes equations. The Quantum FNO achieves similar accuracy with less amount of parameters. It prepares scientific computing for near term quantum device.

**Summary Of The Review:**

Overall, it's good to see the connection. However, it seems the real application is still quite distant. The review feels the paper is marginally below the threshold of acceptance.

---

> ### Author Response · Authors · 2022-11-16
> **Reply to reviewer's comments and concerns.**
>
> We thank the reviewer for the constructive review. Below we provide answers to raised concerns.
>
> 1. The Quantum FFT seems not to be justified. Unnecessary to use quantum language when everything is classical.
>
> We appreciate this concern, but the concept of Quantum Fourier Transform is one of the most important algorithms in Quantum Computing, showing a provable speed up over classical FFT, and at the core of Shor’s famous algorithm for prime numbers. In this paper, we propose a new type of Quantum Fourier Transform, tailored for unary data loading. Acting on vector of size n, the runtime of our Quantum Fourier Transforms scales as O(log(n)), whereas the best classical FFT scales as O(nlog(n)). Therefore we deny that “everything is classical” in this work. If the reviewer was mentioning the numerical experiments made solely on classical computer, this is justified by the lack of available good quality quantum computers. However, it is designed keeping future progress in mind.
>
> 2. Empirically, in classical FNO, the FFT is already quite fast and the matrix multiplication takes the most computation since the channel dimension of FNO can be very large.
>
> In fact, our quantum FNO proposes to speedup both the FFT and the matrix multiplication. Note that in the classical paper, it is highlighted that the classical operation is bounded by complexity of Fourier transform and the classical paper also assumes number of samples to be way larger than number of channels. Also, the matrix multiplication can be parallelised to make it faster. This can be verified experimentally as well. Both in the implementation and in the paper, the number of channels are in range (32-60), whereas the number of samples would be very high (8096) even after subsampling(1024). This is why the FFT becomes a bottleneck operation and where quantum computing can save significant time. Even if it were not the case, we provide faster method for doing matrix multiplication.
>
> 3. It's not very sure if quantum FNO can be more efficient in practice.
>
> Given the progress of development in quantum hardware, it might be usable in the near future and can save a lot of time when deployed for more complex PDEs requiring a large amount of samples. There is a polynomial speedup there which can be exploited on the real quantum computers.
>
>
> 4. The Mnist example looks unnecessary. There exists a previous work on Imagenet.
>
> The reviewer’s concern is shared by the authors, and we wish to have more image classification examples in the future (like ImageNet). We have provided the MNIST example only as a baseline for future work. In addition, we want to emphasize that this work primarily targets PDE solving, whereas image classification is just a side, interesting, application. Some claims have been rephrased in the paper to make this more explicit.
>
> 5. There exists a previous work that uses matrix butterfly decomposition for neural operators
>
> Thanks for pointing this out, we have acknowledged it in the paper and compare the functioning with it in the appendix. Any classical work showing that butterfly matrices are good weight matrices is a plus for our work. We note that no previous work used these matrices in the context of Fourier Neural Operator, or in Quantum Computing. Moreover, our work does not use “butterfly” for the matrices, but for the architecture which is more closely related to the architecture of the Cooley-Tukey FFT algorithm.

---

### Decision · Program_Chairs · 2023-01-20

**Decision:**

Reject

**Justification For Why Not Higher Score:**

There are clear concerns from the reviewers, such as
•	The motivation of the work needs to be further enhanced, including the use of quantum language in this context.
•	Empirically, the performance of the proposed method seems not very good, although the speed is assumed to be increased (the real speed cannot be directly measured given the unavailability of quantum computer).
•	There are some technical errors in the paper, as pointed out by one of our reviews, although the authors have tried to address them in their revisions.
•	The technical novelty of the work is limited, given some highly related previous work.

**Justification For Why Not Lower Score:**

N/A

**Metareview: Summary, Strengths And Weaknesses:**

In this paper, the authors extend the classical Fourier neural operator with Quantum circuits. It applies butterfly circuits as a model compression. The paper studies Burgers, Darcy, and Navier-Stokes equations. While the work is interesting, and might prepare scientific computing for future scientific computing, the reviewers believe its quality has not reached the bar of ICLR. Some reasons are listed below:
•	The motivation of the work needs to be further enhanced, including the use of quantum language in this context.
•	Empirically, the performance of the proposed method seems not very good, although the speed is assumed to be increased (the real speed cannot be directly measured given the unavailability of quantum computer).
•	There are some technical errors in the paper, as pointed out by one of our reviews, although the authors have tried to address them in their revisions.
•	The technical novelty of the work is limited, given some highly related previous work.